# SEMANTICALLY-ADAPTIVE UPSAMPLING FOR LAYOUT-TO-IMAGE TRANSLATION

## ABSTRACT

We propose the Semantically-Adaptive UpSampling (SA-UpSample), a general and highly effective upsampling method for the layout-to-image translation task. SA-UpSample has three advantages: 1) Global view. Unlike traditional upsampling methods (e.g., Nearest-neighbor) that only exploit local neighborhoods, SA-UpSample can aggregate semantic information in a global view. 2) Semantically adaptive. Instead of using a fixed kernel for all locations (e.g., Deconvolution), SA-UpSample enables semantic class-specific upsampling via generating adaptive kernels for different locations. 3) Efficient. Unlike Spatial Attention which uses a fully-connected strategy to connect all the pixels, SA-UpSample only considers the most relevant pixels, introducing little computational overhead. We observe that SA-UpSample achieves consistent and substantial gains on six popular datasets. The source code will be made publicly available.

## 1 INTRODUCTION

The layout-to-image translation task aims to translate input layouts to realistic images (see Fig. 1(a)), which have many real-world applications and draw much attention from the community (Park et al., 2019; Liu et al., 2019; Jiang et al., 2020; Tang et al., 2020). For example, Park et al. (2019) propose GauGAN with a novel spatially-adaptive normalization to generate realistic images from semantic layouts. Liu et al. (2019) propose CC-FPSE, which predicts convolutional kernels conditioned on the semantic layout and then generate the images. Tang et al. (2020) propose LGGAN with several local generators for generating realistic small objects. Despite the interesting exploration of these methods, we can still observe artifacts and blurriness in their generated images because they always adopt the nearest-neighbor interpolation to upsample feature maps and then to generate final results.

Feature upsampling is a key operation in the layout-to-image translation task. Traditional upsampling methods such as nearest-neighbor, bilinear, and bicubic only consider sub-pixel neighborhood (indicated by white circles in Fig. 1(b)), failing to capture the complete semantic information, e.g., the head and body of the dog, and the front part of the car. Learnable upsampling methods such as Deconvolution (Noh et al., 2015) and Pixel Shuffle Shi et al. (2016) are able to obtain the global information with larger kernel size, but learns the same kernel (indicated by the white arrows in Fig. 1(c)) across the image, regardless of the semantic information. Other feature enhancement methods such as Spatial Attention (Fu et al., 2019) can learn different kernels (indicated by different color arrows in Fig. 1(d)), but it still inevitable captures a lot of redundant information, i.e., 'grasses' and 'soil'. Also, it is prohibitively expensive since it needs to consider all the pixels.

To fix these limitations, we propose a novel Semantically-Adaptive UpSampling (SA-UpSample) for this challenging task in Fig. 1(e). Our SA-UpSample dynamically upsamples a small subset of relevant pixels based on the semantic information, i.e., the green and the tangerine circles represent the pixels within the dog and the car, respectively. In this way, SA-UpSample is more efficient than both Deconvolution, Pixel Shuffle, and Spatial Attention, and can capture more complete semantic information than traditional upsampling methods such as the nearest-neighbor interpolation.

We perform extensive experiments on six popular datasets with diverse scenarios and different image resolutions, i.e., Cityscapes (Cordts et al., 2016), ADE20K (Zhou et al., 2017), COCO-Stuff (Caesar et al., 2018), DeepFashion (Liu et al., 2016), CelebAMask-HQ (Lee et al., 2020), and Facades (Tyleček & Šára, 2013). We show that with the help of SA-UpSample, our framework can synthesize better results compared to several state-of-the-art methods. Moreover, an extensive ablation

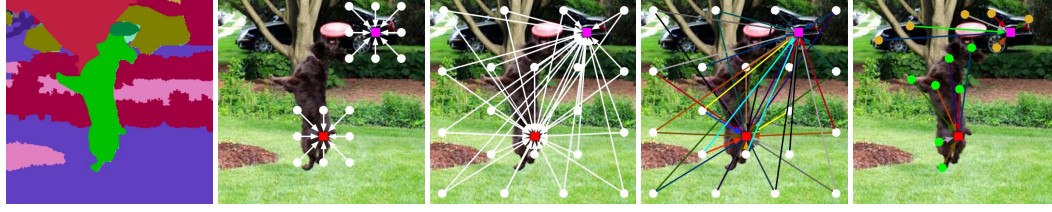

(a) Input Layout   (b) Nearest, etc.   (c) Deconvolution   (d) Spatial Attention   (e) SA-UpSample

Figure 1: Comparison with different feature upsampling and enhancement methods on the layout-to-image translation task. Given two locations $l'$ (indicated by red and megenta squares) in the output feature map $f'$, our goal is to generate these locations by selectively upsampling several points $N(l, k)$ (indicated by circles) in the input feature map $f$.

study shows the effectiveness of SA-UpSample against other feature upsampling and enhancement methods for the layout-to-image translation task.

## 2   RELATED WORK

**Feature Upsampling.** Traditional upsampling methods such as nearest-neighbor and bilinear interpolations use spatial distance and hand-crafted kernels to capture the correlations between pixels. Recently, several deep learning methods such as Deconvolution (Noh et al., 2015) and Pixel Shuffle Shi et al. (2016) are proposed to upsample feature maps using learnable kernels. However, these methods either exploit semantic information in a small neighborhood or use a fixed kernel. Some other works of super-resolution, inpainting, denoising (Mildenhall et al., 2018; Wang et al., 2019; Jo et al., 2018; Hu et al., 2019) also explore using learnable kernels. However, the settings of these tasks are significantly different from ours, making their methods cannot be used directly.

**Layout-to-Image Translation** tries to convert semantic layouts into realistic images (Park et al., 2019; Liu et al., 2019; Jiang et al., 2020; Tang et al., 2020; Zhu et al., 2020a; Ntavelis et al., 2020; Zhu et al., 2020b). Although existing methods have generated good images, we still see unsatisfactory aspects mainly in the generated content details and intra-object completions, which we believe is mainly due to they always adopt the nearest-neighbor interpolation to upsample feature maps and then generate final results. To fix this limitation, we propose a novel Semantically-Adaptive Up-Sampling (SA-UpSample) for this task. To the best of our knowledge, we are the first to investigate the influence of feature upsampling on this challenging task.

## 3   SEMANTICALLY-ADAPTIVE UPSAMPLING (SA-UPSAMPLE)

An illustration of the proposed Semantically-Adaptive UpSampling (SA-UpSample) is shown in Fig. 2, which mainly consists of two branches, i.e., the Semantically-Adaptive Kernel Generation (SAKG) branch predicting upsample kernels according to the semantic information, and the Semantically-Adaptive Feature Upsampling (SAFU) branch selectively performs the feature upsampling based on the kernels learned in SAKG. All components are trained in an end-to-end fashion so that the two branches can benefit from each other.

Specifically, given a feature map $f \in \mathbb{R}^{C \times H \times W}$ and an upsample scale $s$, SA-UpSample aims to produce a new feature map $f' \in \mathbb{R}^{C \times Hs \times Ws}$. For any target location $l' = (i', j')$ in the output $f'$, there is a corresponding source location $l = (i, j)$ at the input $f$, where $i = \lfloor i'/s \rfloor$, $j = \lfloor j'/s \rfloor$. We denote $N(l, k)$ as the $k \times k$ sub-region of $f$ centered at the location $l$ in, i.e., the neighbor of the location $l$. See Fig. 1 and 2 for illustration.

### 3.1   SEMANTICALLY-ADAPTIVE KERNEL GENERATION (SAKG) BRANCH

This branch aims to generate a semantically-adaptive kernel at each location according to the semantic information, which consists of four modules, i.e., Feature Channel Compression, Semantic Kernel Generation, Feature Shuffle, and Channel-wise Normalization.

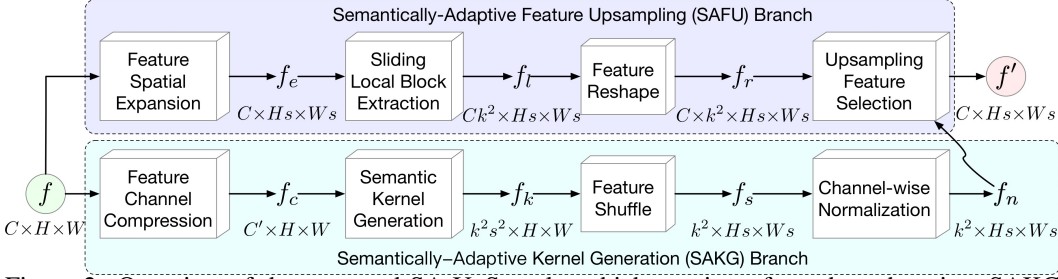

Figure 2: Overview of the proposed SA-UpSample, which consists of two branches, i.e., SAKG and SAFU. SAKG branch aims to generate semantically-adaptive kernels according to the input layout. SAFU branch aims to selectively upsample the feature $f \in C \times H \times W$ to the target one $f' \in C \times Hs \times Ws$ based on the kernels learned in SAKG, where $s$ is the expected upsample scale.

**Feature Channel Compression.** This module is used to reduce the network parameters and computational cost. Specifically, the input feature $f$ is fed into a convolution layer with $1 \times 1$ kernel to compress the input channel from $C$ to $C'$, making SA-UpSample with fewer network parameters.

**Semantic Kernel Generation.** This module receives the feature $f_c \in \mathbb{R}^{C' \times H \times W}$ as input (where $H$ and $W$ denotes the height and width of the feature map) and ties to generate different semantically-adaptive kernels, which can be represented as $f_k \in \mathbb{R}^{k^2 s^2 \times H \times W}$, where $k$ is the semantically-adaptive upsampling kernel size and $s$ is the expected upsample scale.

**Feature Shuffle.** We then feed the feature $f_k$ through a feature shuffle layer for rearranging elements in $f_k$, leading to a new feature map $f_s \in \mathbb{R}^{k^2 \times Hs \times Ws}$, where $k^2 = k \times k$ represents the learned semantic kernel. Note that the learned semantic kernels are quit different at different locations $l'$, as shown in Fig. 1 and 3.

**Channel-wise Normalization.** After that, we perform a channel-wise softmax operation on each semantic kernel $f_s$ to obtain the normalized kernel $f_n$, i.e., the sum of the weight values in $k^2$ is equal to 1. In this way, we can guarantee that information from the combination would not explode. Also, the semantically-adaptive kernels can encode where to emphasize or suppress according to the semantic information.

## 3.2 SEMANTICALLY-ADAPTIVE FEATURE UPSAMPLING (SAFU) BRANCH

This branch aims to upsample the input feature $f$ based on the kernel $f_n$ learned in the SAKG branch in a semantically-adaptive way, which contains four modules, i.e., Feature Spatial Expansion, Sliding Local Block Extraction, Feature Reshape, and Upsampling Feature Selection.

**Feature Spatial Expansion.** The input feature $f$ is fed into this module to expand the size of spatial from $H \times W$ to $Hs \times Ws$. Specifically, we adopt the nearest interpolation to achieve this process.

**Sliding Local Block Extraction.** Then the expanded feature $f_e \in \mathbb{R}^{C \times Hs \times Ws}$ is fed into this module to extract sliding local block of each location in $f_e$, leading to the new feature $f_l \in \mathbb{R}^{Ck^2 \times Hs \times Ws}$.

**Feature Reshape.** Thus, we reshape $f_l$ by adding a dimension, resulting in a new feature $f_r \in \mathbb{R}^{C \times k^2 \times Hs \times Ws}$. In this way, we can do multiplication between the reshaped local block $f_r$ and the learned kernel $f_n$.

**Upsamling Feature Selection.** Finally, the feature map $f_r$ and the kernel $f_n$ learned in the SAKG branch are fed into the Upsampling Feature Selection module to generate the final feature map $f' \in \mathbb{R}^{C \times Hs \times Ws}$ by an weighted sum manner. The computation process at the location $l = (i, j)$ can be expressed as follow,

$$f' = \sum_{p=-\lfloor k/2 \rfloor}^{\lfloor k/2 \rfloor} \sum_{q=-\lfloor k/2 \rfloor}^{\lfloor k/2 \rfloor} f_r(i+p, j+q) \times f_n(p, q). \tag{1}$$

In this way, the pixels in the learned kernel $f_n$ contributes to the upsampled pixel $l'$ differently, based on the semantic information of features instead of the spatial distance of locations. The semantics of the upsampled feature map can be stronger than the original one, since the information from

Input Generated Image Learned Semantically-Adaptive Kernels

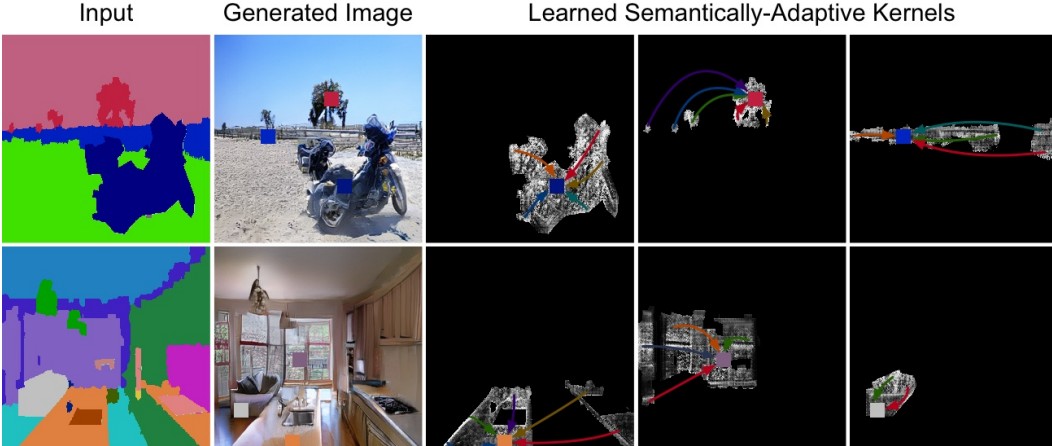

Figure 3: Visualization of learned semantically-adaptive kernels on COCO-Stuff. In each row, we show three representative locations with different color squares in the generated image. The other three images are learned semantically-adaptive kernels for those three locations, with corresponding color arrows summarizing the most-attended regions for upsampling the target location. We observe that the network learns to allocate attention according to regions within the same semantic information, rather than just spatial adjacency.

relevant points in a local region can be more attended, and the pixels with the same semantic label can achieve mutual gains, improving intra-object semantic consistency.

### 3.3 WHY DOES THE SA-UPSAMPLE WORK BETTER?

A short answer is that it can better preserve semantic information against common upsampling methods. Specifically, while other upsampling methods such as nearest-neighbor interpolation and Deconvolution are essential parts in almost all the state-of-the-art image generation (Radford et al., 2016) and translation (Park et al., 2019) models, they tend to 'pollute' semantic information when performing feature upsampling since it would inevitably incorporate contaminating information from irrelevant regions (see Fig. 1).

In contrast, the proposed SA-UpSample performs feature upsampling by using itself, i.e., it uses the pixels belonging to the same semantic label to upsample the feature maps. Hence, the generator can better preserve semantic information. It enjoys the benefit of feature upsampling without losing the input semantic information. In Fig. 3, we show some examples of the learned semantically-adaptive kernels. We can easily observe that the proposed SA-UpSample upsamples features by leveraging complementary features in the regions of the same semantic information than local regions of fixed shape to generate consistent objects/scenarios, further confirming our motivations.

### 3.4 OPTIMIZATION OBJECTIVE AND TRAINING DETAILS

We follow GauGAN (Park et al., 2019) and use three losses as our training objective, i.e., $\mathcal{L}=\lambda_{gan}\mathcal{L}_{gan}+\lambda_f\mathcal{L}_f+\lambda_p\mathcal{L}_p$, where $\mathcal{L}_{gan}$, $\mathcal{L}_f$ and $\mathcal{L}_p$ represent adversarial, discriminator feature matching, and perceptual loss, respectively. We set $\lambda_{gan}=1$, $\lambda_f=10$, and $\lambda_p=10$ in our experiments. We use the multi-scale discriminator (Park et al., 2019) as our discriminator. Moreover, we set $C'=64$, $k=5$ and $s=2$ in the proposed SA-UpSample. We replace the upsampling layers in GauGAN with our SA-UpSample layers. The proposed method is implemented by using PyTorch (Paszke et al., 2019). We conduct the experiments on NVIDIA DGX1 with 8 32GB V100 GPUs.

## 4 EXPERIMENTS

**Datasets.** We first follow GauGAN (Park et al., 2019) and conduct experiments on Cityscapes (Cordts et al., 2016), ADE20K (Zhou et al., 2017), and COCO-Stuff (Caesar et al., 2018). Then we conduct experiments on three more datasets with diverse scenarios. 1) Facades (Tyleček & Šára, 2013) contains different city images with diverse architectural styles. The training and test set

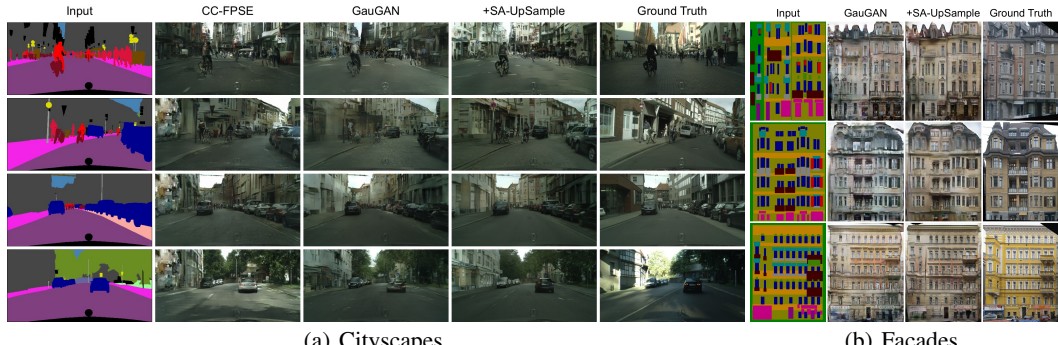

Figure 4: Qualitative comparison on Cityscapes and Facades. From left to right: Input, CC-FPSE (Liu et al., 2019), GauGAN (Park et al., 2019), GauGAN+SA-UpSample (Ours), and GT.

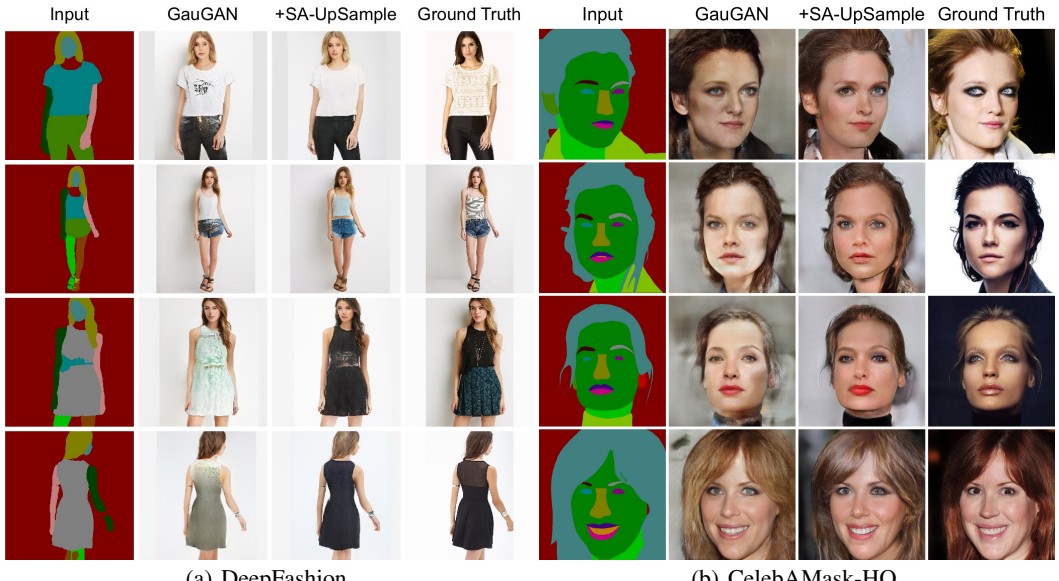

Figure 5: Qualitative comparison on DeepFashion and CelebAMask-HQ. From left to right: Input, GauGAN (Park et al., 2019), GauGAN+SA-UpSample (Ours), and GT.

Table 1: User study. The numbers indicate the percentage of users who favor the results of our method (i.e., GauGAN+SA-UpSample) over GauGAN.

| AMT ↑ | Cityscapes | ADE20K | COCO-Stuff | DeepFashion | Facades | CelebAMask-HQ |
|---|---|---|---|---|---|---|
| Ours vs. GauGAN | 63.8 | 65.7 | 62.4 | 60.1 | 58.3 | 70.5 |

sizes are 378 and 228, respectively. We resize the images to $512{\times}512$ for high-resolution layout-to-image translation tasks. 2) CelebAMask-HQ (Lee et al., 2020) contains face images with 19 semantic facial attributes. The training and test set sizes are 24,183 and 2,842, respectively. We also resize the images to $512{\times}512$. 3) DeepFashion (Liu et al., 2016) contains human body images. The training and test set sizes are 30,000 and 2,247, respectively. We resize the images to $256{\times}256$, and use a well-trained model (Li et al., 2019) to extract input semantic layouts for this dataset.

**Evaluation Metrics.** We follow GauGAN (Park et al., 2019) and use mean Intersection-over-Union (mIoU), pixel accuracy (Acc), and Fréchet Inception Distance (FID) (Heusel et al., 2017) as the evaluation metrics on Citysacpes, ADE20K, and COCO-Stuff. For DeepFashion, CelebAMask-HQ, and Facades, we use FID and LPIPS (Zhang et al., 2018) as the evaluation metrics.

## 4.1 COMPARISONS WITH STATE-OF-THE-ART

**Qualitative Comparisons.** We first compare SA-UpSample with GauGAN (Park et al., 2019) on DeepFashion, CelebAMask-HQ, and Facades. Note that we used the source code provided by the

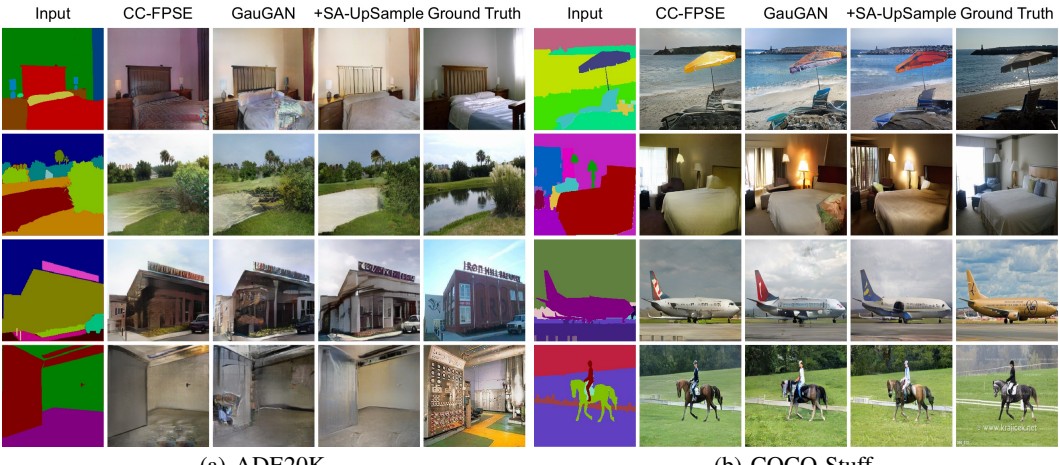

(a) ADE20K.                                                    (b) COCO-Stuff.

Figure 6: Qualitative comparison on ADE20K and COCO-Stuff. From left to right: Input, CC-FPSE (Liu et al., 2019), GauGAN (Park et al., 2019), GauGAN+SA-UpSample (Ours), and GT.

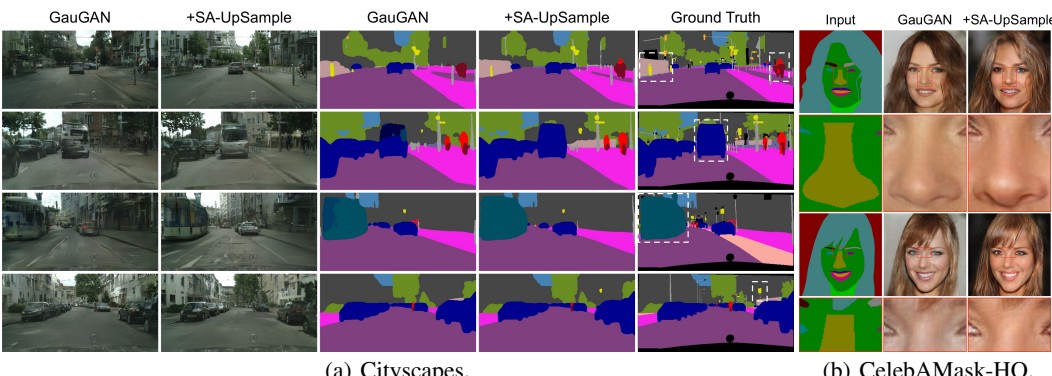

(a) Cityscapes.                                               (b) CelebAMask-HQ.

Figure 7: (a) Visualization of generated semantic maps compared with those from GauGAN (Park et al., 2019) on Cityscapes. Most improved regions are highlighted in the ground truths with white dash boxes. (b) Comparison in a zoomed-in manner on CelebAMask-HQ.

Table 2: Quantitative comparison on DeepFashion, Facades, and CelebAMask-HQ.

| Method | DeepFashion | | Facades | | CelebAMask-HQ | |
|---|---|---|---|---|---|---|
| | FID ↓ | LPIPS ↓ | FID ↓ | LPIPS ↓ | FID ↓ | LPIPS ↓ |
| GauGAN (Park et al., 2019) | 22.8 | 0.2476 | 116.8 | 0.5437 | 42.2 | 0.4870 |
| +SA-UpSample (Ours) | **20.8** | **0.2446** | **112.4** | **0.5387** | **33.6** | **0.4788** |

authors to generate the results of GauGAN on these three datasets for fair comparisons. Visualization results are shown in Fig. 4(b) and 5. We can see that the model with SA-UpSample generates more photo-realistic results than the original GauGAN. Moreover, we compare GauGAN and the proposed method in a zoomed-in manner on CelebAMask-HQ in Fig. 7(b). We can see that the model with our SA-UpSample can generate more vivid content than the original GauGAN model, further validating the effectiveness of SA-UpSample. Lastly, we compare the proposed method with GauGAN and CC-FPSE (Liu et al., 2019) on Cityscapes, ADE20K, and COCO-Stuff. Comparison results are shown in Fig. 4(a) and 6. We can see that our method produces more clear and visually plausible results than both leading methods, further demonstrating our design motivation.

**User Study.** We follow the same evaluation protocol of GauGAN and also perform a user study. The results compared with the original GauGAN are shown in Table 1. We see that users strongly favor the results generated by our proposed method on all datasets, further validating that the generated images by our upsampling method are more photo-realistic.

**Quantitative Comparisons.** Although the user study is more suitable for evaluating the quality of the generated images, we also follow GauGAN and use mIoU, Acc, FID, and LPIPS for quantitative

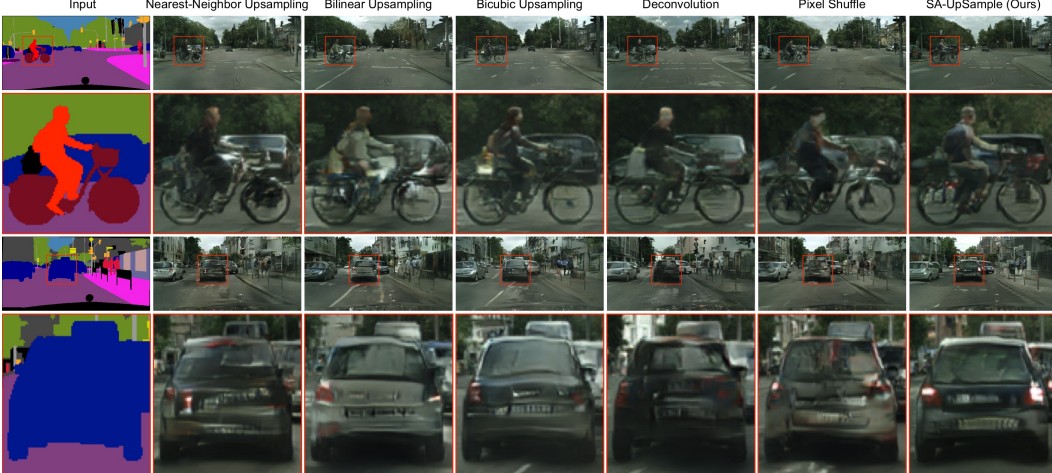

| Input | Nearest-Neighbor Upsampling | Bilinear Upsampling | Bicubic Upsampling | Deconvolution | Pixel Shuffle | SA-UpSample (Ours) |

Figure 8: Qualitative comparison of different upsampling methods on Cityscapes. Key differences are highlighted by red boxes.

Table 3: Quantitative comparison on Cityscapes, ADE20K, and COCO-Stuff. The results of other methods are reported from their papers.

| Method | Cityscapes | | | ADE20K | | | COCO-Stuff | | |
|---|---|---|---|---|---|---|---|---|---|
| | mIoU ↑ | Acc ↑ | FID ↓ | mIoU ↑ | Acc ↑ | FID ↓ | mIoU ↑ | Acc ↑ | FID ↓ |
| CRN (Chen & Koltun, 2017) | 52.4 | 77.1 | 104.7 | 22.4 | 68.8 | 73.3 | 23.7 | 40.4 | 70.4 |
| SIMS (Qi et al., 2018) | 47.2 | 75.5 | 49.7 | - | - | - | - | - | - |
| Pix2pixHD (Wang et al., 2018) | 58.3 | 81.4 | 95.0 | 20.3 | 69.2 | 81.8 | 14.6 | 45.7 | 111.5 |
| CC-FPSE (Liu et al., 2019) | 65.5 | 82.3 | 54.3 | 43.7 | 82.9 | 31.7 | 41.6 | 70.7 | 19.2 |
| PIS (Dundar et al., 2020) | 64.8 | 82.4 | 96.4 | - | - | - | 38.6 | 69.0 | 28.8 |
| TSIT (Jiang et al., 2020) | 65.9 | 82.7 | 59.2 | 38.6 | 80.8 | 31.6 | - | - | - |
| LGGAN (Tang et al., 2020) | 68.4 | 83.0 | 57.7 | 41.6 | 81.8 | 31.6 | - | - | - |
| GauGAN (Park et al., 2019) | 62.3 | 81.9 | 71.8 | 38.5 | 79.9 | 33.9 | 37.4 | 67.9 | 22.6 |
| +SA-UpSample (Ours) | **65.5** | **82.5** | **48.3** | **39.8** | **80.7** | **32.0** | **39.0** | **69.1** | **20.1** |

evaluation. The results compared with several leading methods are shown in Tables 2 and 3. Firstly, we observe that the model with SA-UpSample achieves better results compared with GauGAN on DeepFashion, CelebAMask-HQ, and Facades in Table 2. Moreover, we can see that our method (i.e., GauGAN+SA-UpSample) achieves competitive results compared with other leading methods on Cityscapes, ADE20K, and COCO-Stuff in Table 3. Notably, LGGAN (Tang et al., 2020) achieves better results than our method, however, it trains each local generator for each semantic class, leading to more parameters and the training time will be significantly increased when increasing the number of semantic classes.

**Visualization of Generated Semantic Maps.** We follow GauGAN and adopt the pretrained DRN-D-105 (Yu et al., 2017) on the generated Cityscapes images to produce semantic maps. The results compared with those produced by GauGAN are shown in Fig. 7(a). We see that the method with our proposed SA-UpSample generates more semantically-consistent results than the original GauGAN.

## 4.2 ABLATION STUDIES

**Baselines.** We conduct an extensive ablation study on Cityscapes to evaluate the effectiveness of the proposed SA-UpSample. As shown in Table 4, B1, B2, and B3 are three traditional upsampling methods. B4 and B5 are two learnable upsampling methods. B6 is the Spatial Attention module proposed in (Fu et al., 2019). B7 is our proposed SA-UpSample.

**Ablation Analysis.** We first compare the proposed SA-UpSample with different upsampling strategies (i.e., B1-B5). The results of the ablation study are shown in Table 4 and Fig. 8. We can see that the proposed SA-UpSample achieves significantly better FID than other upsampling methods in Table 4, indicating that the design of effective upsampling methods is critical for this challenging task. We also observe that the proposed SA-UpSample generates more photo-realistic and semantically-consistent results with fewer artifacts than other upsampling methods in Fig. 8. Moreover, we add

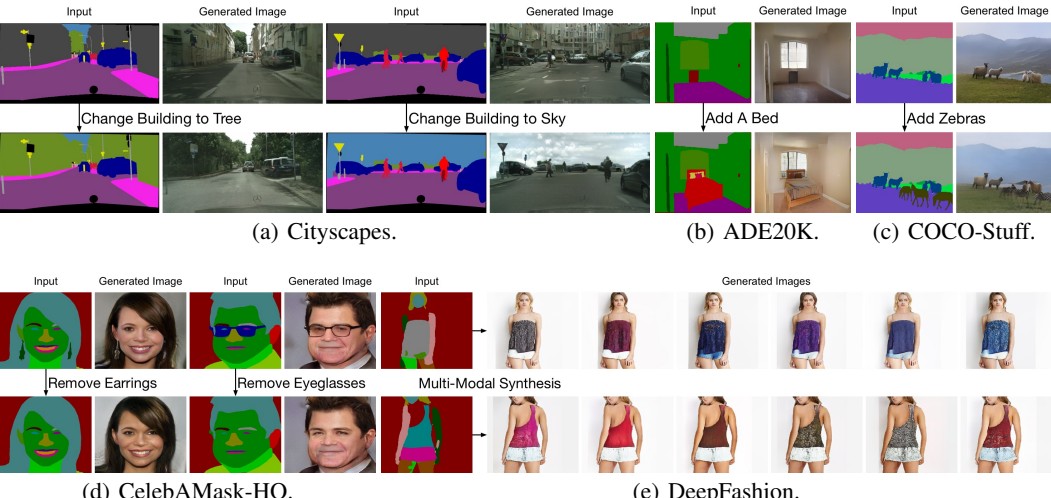

(a) Cityscapes.         (b) ADE20K.    (c) COCO-Stuff.

(d) CelebAMask-HQ.           (e) DeepFashion.

Figure 9: Exemplar applications of the proposed method on different datasets.

the Spatial Attention module (Fu et al., 2019) to GauGAN, obtaining 56.2 on FID. We can see that our method still significantly outperforms Spatial Attention.

**Comparisons of Model Parameters.** We also compare the number of generator parameters with different baselines. The results are shown in Table 4. Traditional upsampling methods B1-B3) have the same number of parameters. Also, we can see that the proposed SA-UpSample achieves superior model capacity than the learnable upsampling methods (i.e., B4 and B5) and Spatial Attention.

Table 4: Quantitative comparison of different feature upsampling and enhancement methods on Cityscapes.

| No. | Method | FID ↓ | Params ↓ |
|-----|--------|-------|----------|
| B1 | Nearest-Neighbor Upsampling | 58.7 | 93.0M |
| B2 | Bilinear Upsampling | 52.9 | 93.0M |
| B3 | Bicubic Upsampling | 54.4 | 93.0M |
| B4 | Deconvolution (Noh et al., 2015) | 54.0 | 98.6M |
| B5 | Pixel Shuffle (Shi et al., 2016) | 59.1 | 143.2M |
| B6 | Spatial Attention (Fu et al., 2019) | 56.2 | 97.4M |
| B7 | SA-UpSample (Ours) | **48.3** | **93.4M** |

**Generalization of SA-UpSample.** Our SA-UpSample is general and can be seamlessly integrated into existing GANs. Thus, to validate the generalization ability of SA-UpSample, we further conduct more experiments on Facades. Specifically, we replace the upsampling layers in CC-FPSE (Liu et al., 2019) with our SA-UpSample. We observe that the CC-FPSE model with SA-UpSample further decreases FID from 93.98 to 90.49, validating the generalization ability of SA-UpSample.

## 4.3 APPLICATIONS

**Semantic Manipulation.** Our model also supports semantic manipulation. For instance, we can replace the building with trees in Fig. 9(a), insert a bed in the room in Fig. 9(b), add a few zebras to the grass in Fig. 9(c), or remove earrings and eyeglasses from faces in Fig. 9(d). These applications provide users more controllability during the translation process.

**Multi-Modal Synthesis.** By using a random vector as the input of the generator, our model can perform multi-modal synthesis (Zhu et al., 2017). The results are shown in Fig. 9(e). We can see that our model generates different outputs from the same input layout.

## 5 CONCLUSION

We present a novel Semantically-Adaptive UpSampling (SA-UpSample), a highly effective upsampling method for the layout-to-image translation task. We observe that SA-UpSample consistently boosts the generation performances on six datasets with diverse scenarios. More importantly, SA-UpSample introduces little computational overhead and can be readily plugged into existing GAN architectures to solve other tasks.

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
