# OpenReview forum: "Semantically-Adaptive Upsampling for Layout-to-Image Translation"
_ICLR.cc/2021/Conference — Reject_

### Official Review · AnonReviewer2 · 2020-10-28
**Novelty is limited**

**Rating:** 5
**Confidence:** 4

**Review:**

This paper presents a module named semantically-adaptive upsampling (SA-UpSample) to achieve layout-to-image translation. The proposed method is able to aggregate semantic information in the layout input and adaptively conducts class-specific upsampling in the translation process. Experiments on six datasets demonstrate the effectiveness of the proposed mehtod. However, there are still several issues to be addressed.

1. Novelty. The proposed semantically-adaptive kernel generation branch and semantically-adpative feature upsampling branch share very similar idea as the filter generation network and dynamic filtering layer in [a], and adaptive interpolation kernel generation and adaptive image resampling in [b], except that the input of generation branch is layout map.

[a] B. De Brabandere et al., Dynamic Filter Networks, NIPS 2016.

[b] X. Jia et al., Super-Resolution with Deep Adaptive Image Resampling, CoRR abs/1712.06463, 2017

2. What if applying the proposed module to also the place after conv layers that corresponds to 1x upsampling? Will performance be further improved.

3. Given the experiments in Fig. 5 (comparing 3rd column and the 4th column), I wonder whether there is any overfitting with the results.

4. It only shows that it is effective in image translation but does not mention how efficient the proposed module is.

5. In Table 3, it seems that the semantic consistency with layout input is worse than TSIT and LGGAN. It is not clear why the row of the proposed method is marked bold although it is not the best in terms of some metrics.

---

### Official Review · AnonReviewer1 · 2020-10-28
**Clarification and comparison with prior art needed**

**Rating:** 5
**Confidence:** 5

**Review:**

**Summary**
This paper presents a content-aware upsampling approach for layout-to-image translation problems. The core idea is to learn a spatially varying upsampling kernel. The paper applied the proposed upsampling method to GauGAN and showed improved performance over the nearest neighbor upsampling operators on six layout-to-image datasets.

**Strength**
+ The results clearly show that the proposed upsampling operator can improve the visual quality over the baseline nearest neighbor upsampling. The quantitative results also support the effectiveness of the proposed upsampling operation.
+ The motivation and the differences over alternative methods is clear (Figure 1).
+ The overview figure helps readers understand the high-level ideas.

**Weakness**

== Exposition ==

The exposition of the proposed method can be improved.
For examples,
-	it’s unclear how the “Semantic Kernel Generation” is implemented. I can probably guess this step is essentially a 1 x 1 convolution, but it would be better to fill in the details.
-	In the introduction, the paper mentioned that the translated images often contain artifacts due to the nearest-neighbor upsampling. Yet, in the Feature Spatial Expansion step it also used nearest-neighbor interpolation. This needs some more justification.
-	For Figure 3, it’s unclear what the learned semantically adaptive kernel visualization means. At each upsampling layer, the kernel is only K x K.
-	At the end of Section 2, I do not understand what it means by “the semantics of the upsampled feature map can be stronger than the original one”.
-	The proposed upsampling layer is called “semantically aware”. However, I do not see anything that’s related to the semantics other than the fact that the input is a semantic map. I would suggest that this should be called “content aware” instead.

== Technical novelty ==
-	My main concern about the paper lies in its technical novelty. There have been multiple papers that proposed content aware filter. As far as I know, the first one is [Xu et al. 2016].
Jia, Xu, et al. "Dynamic filter networks." Advances in neural information processing systems. 2016.

- The work most relevant to this paper is the CARAFE [Wang et al. ICCV 2019], which can be viewed as a special case of the dynamic filter network for feature upsampling. By comparing Figure 2 in this paper and Figure 2 in the CARAFE paper [Wang et al. ICCV 2019], it seems to me that the two methods are * the same*. The only difference is the application of the layout-to-image task. The paper in the related work section suggests, “the settings of these tasks are significantly different from ours, making their methods cannot be used directly.” I respectfully disagree with this statement because both methods take in a feature tensor and produce an upsampled feature tensor. I believe that the application would be straightforward without any modification. Given the similarity to prior work, it would be great for the authors to (1) describe in detail the differences (if any) and (2) compare with prior work that also uses spatially varying upsampling kernel.

Minor:
-	CARAFE: Content-Aware ReAssembly of Features. The paper is in ICCV 2019, not CVPR 2019

In sum, I think the idea of the spatially adaptive upsampling kernel is technically sound. I also like the extensive evaluation in this paper. However, I have concerns about the high degree of similarity with the prior method and the lack of comparison with CARAFE.

** After discussions **

I have read other reviewers' comments. Many of the reviewers share similar concerns regarding the technical novelty of this work. I don't find sufficient ground to recommend acceptance of this paper.

---

### Official Review · AnonReviewer4 · 2020-10-29
**Some details of the proposed method are unclear**

**Rating:** 6
**Confidence:** 5

**Review:**

This paper proposes the Semantic-Adaptive UpSampling method to do feature upsample in layout-to-image translation task. The SA-UpSample module exploit the semantic information to learn adaptive upsample kernels for different input features. The proposed method has the advantage of global view, semantically-adaptive and efficient. Extensive experimental results are shown to prove the effectiveness of the proposed method. The paper is well organized and easy to follow.

Strength:

(+) The experiments are sufficient and extensive. The proposed method is evaluated on many different datasets and generate satisfactory results in all of these datasets. Both qualitative and Quantitative results are shown, along with user study and visualization. Comparison results with different upsampling methods and state-of-the-art methods guarantee the effectiveness of the proposed methods.

(+) This paper is well organized and the description of the proposed method is easy to understand.

Weakness:

(-) How to determine some hyper-parameters of the proposed method is not clear. E.g. C’, k and s are very important hyper-parameters of SA-UpSample. The authors set C’=64, k=5 and s=2. Are they determined by manual or from experimental results?

(-) The proposed method is efficient because it has a feature channel compression step. If we ignore the computational complexity and enlarge the channel number of this step (i.e. enlarge C’), will the proposed method generate better results (images with more realistic)? Whether the authors choose a trade off between effectiveness and efficiency by setting a relatively small C’?

---

### Official Review · AnonReviewer3 · 2020-10-30
**Incremental novelty and limited performance improvement**

**Rating:** 4
**Confidence:** 5

**Review:**

##########################################################################

Summary:

This paper proposes a semantically-adaptive upsampling approach for layout-to-image translation. It uses the semantic label map to predict spatially-adaptive upsampling kernels for feature map upsampling.  Compared with traditional upsampling methods, it has a larger receptive field to focus on not only nearby pixels, but also semantically-related pixels at a longer distance. Experiments are conducted on Cityscapes, ADE20K, COCO-Stuff, DeepFashion, CelebAMask-HQ, and Facades datasets, and the proposed approach achieves better results compared with the baseline.

##########################################################################

Pros:

1. The semantically-adaptive upsampling approach considers the semantic information for upsampling. It has a larger receptive field and can take far-away pixels which are semantically related for upsampling, so that it can better preserve the semantic consistency within the instance or stuff with the same semantic label.

2. The proposed semantically-adaptive upsampling is efficient compared with spatial attention or prediction convolution kernels.

3. The writting and explanations are clear.

##########################################################################

Cons:

1. In Figure.2, both the SAFU branch and the SAKG branch take the feature f as input. Based on my understanding of this paper, the input of the SAKG branch should be the semantic layout map, rather than the feature map f. I think this figure is a little confusing and misleading.

2. The novelty of the semantically-adaptive upsampling is limited. On the one hand, semantic-adaptive operations have been proposed in previous work SPADE (semantically-adaptive normalization) and CC-FPSE (semantically-adaptive convolution). The difference from the conditional convolution in CC-FPSE is that the semantically-adaptive upsampling does not learn the feature transformation across channels, so the parameter size to be predicted is smaller. On the other hand, content-adaptive upsampling operations have also been explored in previous work CARAFE [A]. Especially the architecture design of this paper is almost the same as CARAFE (even with the same submodule names such as "channel compression" and "channel-wise normalization"). The only difference from CARAFE is that the proposed method uses the semantic label maps rather than the feature maps to predict the upsampling kernels.

[A] CARAFE: Content-Aware ReAssembly of FEatures, ICCV 2019

3. The authors claim that the proposed upsampling approach can aggregate information in a global view, but the receptive field of the upsampling layer is still limited to the kernel size of the upsampling kernels. The kernel size used in this paper is k=5. Have the authors experiment using different kernel sizes to see the effect of kernel size on the image generation performance?

4. The visual results do not seem to be significantly better than previous methods.

##########################################################################

Reasons for score:

My main reason for rating this paper as below the acceptance threshold is that the noveltly is limited (see the 2nd point in Cons for explanation), and there is not a large improvement on the quality of the synthesized images.

##########################################################################

Questions during the rebuttal period:

Please address my concerns in the Cons part.

##########################################################################

Post-rebuttal:

I have read other reviewers' comments. Since the authors did not provide feedback to our reviews, I would change my score from 5 to 4.

---

### Decision · Program_Chairs · 2021-01-07
**Final Decision**

**Decision:**

Reject

**Comment:**

The paper proposes an upsampling layer design for converting layouts to images. Three reviewers rate the paper below the bar, while one reviewer rates the paper marginally above the bar. The main concern that several reviewers raise is the novelty. Particularly, R1 and R3 point out that the proposed method shares great similarity to CARAFE [Wang et al. ICCV 2019]. The AC agrees with the reviewers.